# Base Models Beat Aligned Models at Randomness and Creativity

**Peter West**[1,2] **& Christopher Potts**[1]
[1]Stanford University
[2]University of British Columbia

## Abstract

Alignment has quickly become a default ingredient in LLM development, with techniques such as reinforcement learning from human feedback making models act safely, follow instructions, and perform ever-better on complex tasks. While these techniques are certainly useful, we propose that they should not be universally applied and demonstrate a range of tasks on which base language models consistently outperform their popular aligned forms. Particularly, we study tasks that require *unpredictable outputs*, such as random number generation, mixed strategy games (rock-paper-scissors and hide-and-seek), and creative writing. In each case, aligned models tend towards narrow behaviors that result in distinct disadvantages, for instance, preferring to generate "7" over other uniformly random numbers, becoming almost fully predictable in some game states, or prioritizing pleasant writing over creative originality. Across models tested, better performance on common benchmarks tends to correlate with worse performance on our tasks, suggesting an effective trade-off in the required capabilities.

## 1 Introduction

The human editors behind "I am Code" (Katz et al., 2023), a popular book of AI poetry, assert that model-written poems get worse with newer, more aligned models (Kestenbaum, 2024). This trend extends to other capabilities such as world modeling (Li et al., 2024) and output diversity (Murthy et al., 2024; Kirk et al., 2024). The prospect that alignment is actively degrading useful capabilities is highly consequential, as the vast majority of LLM users exclusively interact with public-facing *aligned* models (Anthropic, 2024; OpenAI et al., 2023; Gemini-Team, 2024). Although these techniques, such as reinforcement learning from human feedback (Ouyang et al., 2022), are consistently validated on popular benchmarks (Fourrier et al., 2024), capabilities such as poetry writing or deploying mixed strategies deviate significantly from these evaluations (figure 1).

In this work, we study a family of tasks that capture this deviation, particularly tasks that require unpredictability in models, such as random number generation, mixed-strategy games, and poetry writing. Contrary to typical benchmark tasks which can be solved with a single correct answer, tasks such as random number generation explicitly require a distribution of answers, and the tendencies of aligned models to converge towards specific correct responses (Li et al., 2024) become a drawback.

We broadly find that standard alignment recipes, although useful for common benchmarks, erode performance for our tasks (figure 1). We observe the effects of a cross section of alignment recipes (SFT, DPO, Tulu, Llama-Instruct) on the widely-used Llama-3.1 base model (Dubey et al., 2024), with alignment causing consistent performance drops across random number generation (§2), mixed strategy games (§3) and creative poetry generation (§4). Aligned models pick up recognizable patterns that often hurt performance, for instance, generating "7" over other equally random numbers, a common human bias (Simon, 1971; Trueman, 1979; Muller, 2024). In games, aligned models tend to become significantly more deterministic, and especially more confident after better outcomes (tie or win). Finally, in creative poetry writing, aligned models seem to sacrifice creative originality in favor of pleasantness, as judged by humans.

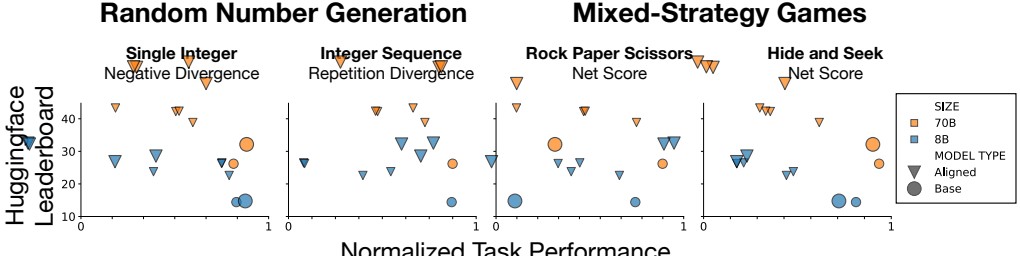

Figure 1: An empirical tradeoff between leaderboard performance and unpredictable capabilities. We compare model performance (8B & 70B parameters) on the popular Open LLM Leaderboard (Fourrier et al., 2024) and various tasks requiring models to be unpredictable, including random number generation and games. Better performance on the leaderboard for aligned models (▼) seems to come at the cost of unpredictable task performance, while Base models (•) do best at these tasks.

Overall, our work provides substantial evidence that popular alignment recipes can reduce capabilities that were present in base models. The unpredictability required in these tasks could be fundamental in developing truly useful agents that are capable of bargaining on our behalf (which requires mixed strategies) or helping humans be more creative. Finally, we note that higher performance on popular benchmarks tends to predict lower performance on our tasks (figure 1), suggesting the possibility of a fundamental trade-off between the sets of skills commonly tested and those required here.

## 2 Basic Randomness: Number Generation

As a basic diagnostic for the ability of models to be unpredictable, we begin with random number generation, in which each generated number should not be predictable beyond random chance. We find that aligned models display significant patterns, resulting in predictability. In generating independent numbers (§2.1.1), even 70B parameter aligned models tend strongly toward generating the number "7", while the base LLMs on which these are built are much closer to uniform. When allowed to generate *sequences* of random numbers (§2.1.2), aligned models become much more uniform in overall frequency, but not necessarily random. For instance, they rarely repeat integers in a given sequence, which improves uniformity but deviates significantly from a truly random process.

### 2.1 Experiments

#### 2.1.1 *Generating Independent Random Numbers*

We begin by testing the ability of models to generate single, uniformly random integers, specifically generating $X \sim \mathcal{U}\{0, 10\}$.

**Setup** We provide brief instructions via zero-shot prompting to each model, following the relevant prompt formatting for the given model (Appendix A.2), with the basic phrasing:

```
Generate a random integer, uniformly between 0 and 10 (inclusive).
```

We use simple rules to parse model outputs, removing any output that does not follow the task specification (a small fraction for all models). For each model, we sample until reaching 1500 successful generations, which we analyze below.

**Models** We hold the base model constant across experiments to one of the most popular options, Llama-3.1 base (Dubey et al., 2024), and investigate the effects of a cross section of 4 strong alignment recipes: Direct Preference Optimization (Tulu-DPO), Supervised Finetuning (Tulu-SFT), the full Tulu-3 instruct recipe (Tulu-Full; Lambert et al. 2025), which combines SFT, DPO, and Reinforcement Learning, and finally the original Meta-Llama 3.1 post-training recipe (Llama-Instruct) which combines DPO, SFT, and rejection sampling. We study two available model sizes, 8B and 70B parameters. For the 70B models, we use FP8

precision to allow these to run locally on 4 NVIDIA RTX A6000 GPUs. Where applicable in this section, "True Sample" indicates the underlying process that we are prompting models to replicate, i.e. `random.randint(0,10)` in Python.

### 2.1.2 Generating Random Number Sequences

To test whether models can account for these biases given their own generation history, we also include a setting in which models attempt to generate length-10 sequences of integers from $X \sim \mathcal{U}\{0, 10\}$.

**Setup** We follow a similar zero-shot prompting format to §2.1.1, simply adding the instruction to generate 10 random numbers instead of one. We use rule-based parsing for these sequences, removing any with $length < 10$ or for which any sequence entries are not integers in $[0, 10]$. We study the same **models** as §2.1.1 here.

## 2.2 Results

In general, we find that these popular alignment recipes seem to *reduce* the randomness of the base model, introducing biases that correspond to common human preferences but deviate from true randomness. Results are in figures 2 and 3 and table 1.

**Alignment increases distributional divergence** Figure 2 shows histograms of single-integer distributions (§2.1.1) across model type. Qualitatively, base models are significantly more uniform across model sizes. One common pattern in aligned models is a tendency to generate "7" with significantly higher probability than other numbers, a common human bias in random numbers (Simon, 1971; Trueman, 1979). While much less dominant, "7" is also the mode of the base distributions, suggesting the bias may begin in the base model and be exacerbated by these alignment recipes. We also include Pearson $\chi^2$ divergence values here (Chernoff & Lehmann, 1954), which are commonly used to measure distance from the uniform distribution. Although the base models (2nd column) deviate significantly more than a true random sample (1st column), aligned models are roughly an order of magnitude worse (or more). Llama-Instruct is the most divergent of the aligned models, generating "7" most of the time, while the supervised finetuned model Tulu-SFT is the least. Note that these issues persist when accounting for entropy using temperature (see Appendix A.3.1)

**Model sequences appear closer to uniform** When we allow models to generate integers in sequence rather than in isolation (§2.1.2), all models become less divergent from the uniform distribution (table 1, full histograms in appendix A.3.1). Some aligned models are less divergent than the base model at 70B parameters, although this only indicates uniformity of frequency in the sequences and not necessarily a more random process. We explore this below.

**Aligned models are biased against repetition** While the overall distribution of integers becomes much more uniform when generating sequences than individual values, this does not necessarily mean that the underlying process is uniformly random. In fact, we find that aligned models follow a human-like heuristic: a tendency away from repeating integers (Wagenaar, 1972; Schulz et al., 2012). While repeated integers may seem *less random* to humans, truly uniform sequences tend to contain them (figure 3). Making sequences non-repeating naturally increases overall uniformity by increasing the coverage of each sequence, but it specifically biases models away from uniform sampling in which probability is independent of previous samples. We compare the number of repetitions for each model and a true uniform sequence ("True Sample") in figure 3, finding that the base model closely resembles the true uniform distribution and all aligned models fundamentally deviate. The most common case for both base models and "True Sample" is sequences with 3 repetitions, while for all aligned models, the mode is zero repetitions. Note that the SFT 70B is closest to having a larger mode, and its mean squared error[1] from the true sample supports the ordering Base $<$ SFT $<$ other aligned models, i.e. base is least divergent from random,

---

[1]Comparing repetition counts against a large-$n$ sampled approximation of expected counts from uniform: $MSE = \left(\frac{1}{11}count_{obs} - count_{expected}\right)^2$ where 11 is the number of bins. The negative of this is used in figure 1.

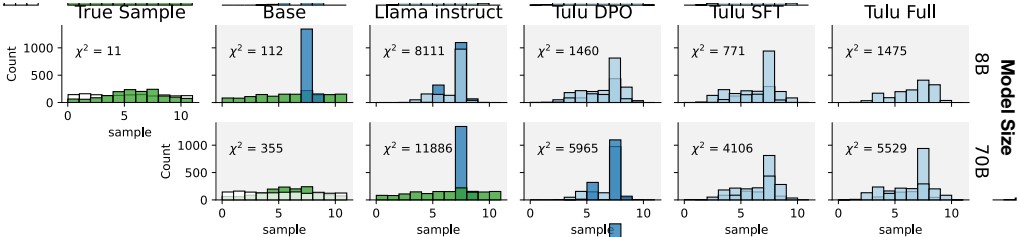

Figure 2: Results of **single value** random number sampling on [0,10] with a **True Sample** using python random.randint(), the Llama-3.1 **Base** (Dubey et al., 2024) model, as well as 4 aligned versions of this same model: **Llama-Instruct** alignment, and 3 kinds of **Tulu** alignment: SFT, DPO, and Full. Aligned models have a consistent preference for "7" compared to the qualitative uniformity of the base model, and significantly higher $\chi^2$ divergence values.

| size | Base | Llama-Instruct | Tulu-DPO | Tulu-SFT | Tulu-Full |
|------|------|----------------|----------|----------|-----------|
| 8 | 13.9 | 115.1 | 100.8 | 52.3 | 129.1 |
| 70 | 29.2 | 43.6 | 22.9 | 21.8 | 18.3 |

Table 1: $\chi^2$ values for sequentially generated random numbers, with sequence lengths of 10 (§2.1.2).

followed by the SFT model and then other aligned models. We include further analysis in Appendix A.3.1 exploring the next-integer distribution of each model over sequence position.

**Scaling laws need not apply** One surprising result here is that issues with randomness do not always disappear or improve with larger model scale. Particularly, in the single-integer generation experiments (§2.1.1), 70B models have a higher divergence than 8B models across the board, including for base LLMs. This disagrees with the general tendency of performance to improve with scale (Kaplan et al., 2020), and suggests that the usefulness of alignment is not the only intuition that may break down in tasks requiring unpredictability. We find similar results in later experiments as well.

## 3 Games Requiring Randomness

Although random number generation (§2) tests the ability of models to be unpredictable, its direct significance is limited given the wealth of existing randomness tools (e.g. the Python random module). Here, we test settings where randomness is required for more complex behavior. Particularly, we study the effects of alignment on **mixed strategy games** (von Neumann & Morgenstern, 1947), where robust strategies must be unpredictable to be robust to deterministic adversaries. §3.1 gives background on mixed strategy games, §3.2 explains the games we test, and §3.3 covers model performance. Broadly, alignment seems to make models less robust to deterministic adversaries, which is in line with our earlier finding of a reduction in randomness (§2).

### 3.1 Background: Mixed Strategy Games

In the context of game theory, pure strategies give a complete, deterministic description of a player's moves. These are a special case of **mixed strategies** (von Neumann & Morgenstern, 1947) which provide a probability distribution over potential pure strategies. In some games, there is no rational pure strategy, i.e. the Nash equilibrium strategy is probabilistic rather than deterministic. Rock Paper Scissors (described in §3.2.1) is an example: if a player uses a pure (deterministic) strategy (e.g. playing "rock" every time), there is an adversarial strategy (playing "paper") which always beats the player.

Mixed strategy games represent a setting in which the failure of models to be random (§2) or unpredictable will explicitly result in negative outcomes. Specifically, models will

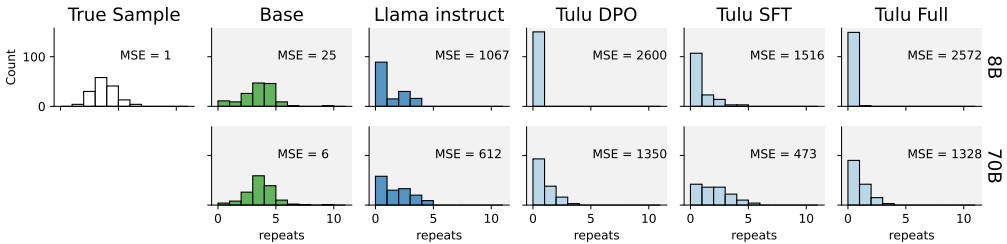

Figure 3: The number of repeated integers in sampled sequences of length 10, as sampled by: **True Sample** using `python random.randint()`, the Llama-3.1 **Base** model and its aligned version using 4 recipes: **Meta-Instruct** alignment, and 3 kinds of **Tulu** alignment: SFT, DPO, and Full. Qualitatively, the base model is by far the closest to the sampled distribution, e.g. the two sizes of the base model are the only models that have the same mode as the true sample of 3 repeated integers. Mean squared error (MSE) is measured against expected counts estimated with a very large empirical sample (10,000 randomly sampled sets).

| Model | Rock Paper Scissors | | | | Hide & Seek | | |
|---|---|---|---|---|---|---|---|
| | Wins | Ties | Losses | Net | Wins | Losses | Net |
| Uniform (limit) | | | | 0.0 | | | 71.4 |
| **8B Parameters** | | | | | | | |
| Base | 26.3 | 25.9 | 47.9 | **-21.6** | 66.9 | 33.1 | **33.8** |
| Llama-Instruct | 20.9 | 21.3 | 57.8 | -36.9 | 43.5 | 56.5 | -13.0 |
| Tulu-DPO | 23.2 | 18.7 | 58.1 | -34.9 | 25.0 | 75.0 | -50.0 |
| Tulu-SFT | 26.0 | 22.5 | 51.4 | -25.4 | 40.9 | 59.1 | -18.2 |
| Tulu-Full | 21.3 | 17.3 | 61.4 | -40.1 | 22.4 | 77.6 | -55.2 |
| **70B Parameters** | | | | | | | |
| Base | 34.4 | 16.2 | 49.4 | **-15.0** | 75.5 | 24.5 | **51.0** |
| Llama-Instruct | 17.0 | 15.9 | 67.1 | -50.1 | 31.0 | 69.0 | -38.0 |
| Tulu-DPO | 21.2 | 23.6 | 55.2 | -34.0 | 34.9 | 65.1 | -30.2 |
| Tulu-SFT | 26.4 | 25.9 | 47.7 | -21.3 | 53.1 | 46.9 | 6.2 |
| Tulu-Full | 22.3 | 21.8 | 55.9 | -33.6 | 33.2 | 66.8 | -33.6 |

Table 2: Outcome rates (%) for different model sizes on Rock Paper Scissors and Hide & Seek, playing against a greedy deterministic adversary with blackbox access to the model. **Best net outcome** for each model size is bolded, second best is underlined.

be vulnerable to deterministic adversaries that have knowledge of the given strategy. In this section, we will test the robustness of each model against such adversaries, assuming knowledge of the underlying move probability of the model.

## 3.2 Experiments

### 3.2.1 Rock Paper Scissors

**Game** Rock Paper Scissors is a multi-round game, with 3 moves (*rock, paper, scissors*) such that *rock beats scissors*, *scissors beat paper*, and *paper beats rock* (while the same move results in a tie). Over multiple rounds, players simultaneously announce moves, accumulating wins, ties, and losses. In our experiments, each model will be playing against a programmatic adversary with knowledge of model probabilities, to test their ability to deploy a mixed strategy.

**Setup** As in §2, we use basic zero-shot prompting to specify the task, keeping language simple to inform the model that it is playing the game, supply any rounds that have been played so far, and ask for the model's next move. Phrasing is consistent across models, besides model-specific formatting. We sample from models based on logit probability, with temperature 1.0 and $\text{top}_k/\text{top}_p$ set to retain the full, original distribution. We then parse outputs to handle formatting that may be included for different models. We need to estimate

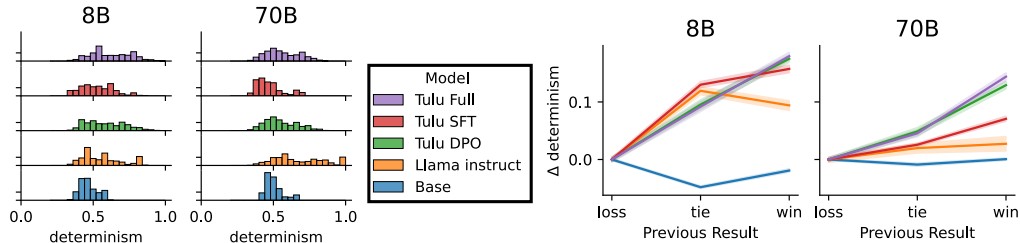

Figure 4: To provide intuition for why aligned models do worse at Rock Paper Scissors, we investigate how *deterministic* models are in each move, using $max_{move} p(move)$ as a measure. Over all rounds (left), we see that all aligned models tested become more deterministic in some rounds than the base models ever do. When plotting how much more deterministic models are after a tie or win vs. a loss (right), we see that the aligned models tend to be more deterministic after a tie or win, while the base models do not consistently show this pattern.

model move probability, both to select a new move for the adversary, and analyze model behavior. To do this, we use the next-token distribution given the prompt $p(t|prompt)$, and aggregate probability across tokens $t$ corresponding to each move, e.g. taking the probability of *rock* to be the combined probability of all tokens that correspond to this move.

We have models play 500 games, with 10 rounds in each game. The set of models is the same as in §2.

**Adversary**  A main feature of mixed strategy games is that unpredictability is required to be robust to deterministic (or *pure*) strategies. If a player is too deterministic, there will be an adversarial strategy that consistently wins against them. Here, we have models compete against deterministic adversaries to test robustness in a mixed strategy setting. An ideal deterministic adversary should take the move at every point that gives the highest expected win rate over the remaining rounds of the game. We apply a greedy approximation for this adversary, using the next-token distribution given the prompt to approximate the model's probability of each next move, and picking the move most likely to counter that (e.g. if we find surface form tokens indicating *rock* add up to 90% of the probability, the adversary would play *paper* next to maximize probability of an immediate win).

### 3.2.2  Hide & Seek

**Game**  We also include **Hide & Seek**, which is an asymmetrical game where one or more players hide, and another player (the *seeker*) attempts to find all other players. We create a simple one-vs-one version where one player (the model) picks a hiding spot every round, and the seeker (adversary) is allowed to choose one location to search. The seeker wins if they pick the same spot, and otherwise loses. This is a mixed strategy game where the equilibrium strategy for a single hiding player is uniformly random. Unlike Rock Paper Scissors the expected result in the equilibrium case is to win $\frac{n-1}{n}$ of the time, when there are $n$ hiding spots. In our experiments, this means that an ideal player will win roughly 85% of the time against an adversarial seeker, i.e. a net score ($rate_{win} - rate_{loss}$) of ≈70 points.

**Setup**  As in §3.2.1, we specify the game simply via zero-shot prompting, informing the model of all hiding spots and asking for a selection in each round, while providing the history of the game so far. We use the same **models** as §3.2.1, and follow the same procedure for the greedy adversary (the seeker), selecting whichever hiding spot the model is most likely to have chosen.

### 3.3  Results

**Base models are consistently most robust against adversaries**  We present results in table 2. Overall, base models achieve the highest scores across games and model sizes, indicating the strongest performance against the adversary. In Rock Paper Scissors, the Tulu-SFT model

achieves the second-best score for both sizes and is within 7 net points in both cases. For Hide & Seek, the base model is at least 40 points above any baseline for both model sizes, with Llama-Instruct 2nd best at 8B parameters and Tulu-SFT second best at 70B parameters, meaning Tulu-SFT is 2nd in 3/4 settings.

**Case Study: Patterns in Determinism for Rock Paper Scissors** To investigate why alignment recipes seem to reduce performance in mixed strategy games, we carry out an in-depth analysis of Rock Paper Scissors. First, we define a measure for how *deterministic* a model is in a given round of a game, as:

$$determinism = max_{move} p(move)$$

In words, this is the probability of the most probable next move for the model to play. The minimum is $determinism = \frac{1}{3}$ for Rock Paper Scissors when models are uniformly random and have the best expected outcome. The maximum is $determinism = 1$ when models are totally deterministic and expected to lose 100% of the time. One interpretation of this score is the degree to which one move is dominating model probability, resulting in behavior more similar to full determinism.

We first investigate the overall distribution of determinism of models across all moves played (i.e. all rounds in all games) in figure 4, left. Base models, which perform best at this game, tend to have determinism near 0.5 and have a very low maximum compared to all other models. For instance, 70B parameter Llama Instruct becomes almost completely deterministic in some cases.

We also find that the result (win, tie, or loss) of the round directly before the given move affects determinism differently in different models (figure 4, right). In this experiment, we plot:

$$mean(determinism|outcome_{i-1}) - mean(determinism|outcome_{i-1} = loss)$$

In words, this is how the outcome of the previous round affects determinism, setting 0 to the case when models lose, to simplify visual comparison. In every case, **aligned models become more confident** after a tie or win than a loss. In contrast, base models are slightly less confident after a tie than a loss, and very similar between a loss and a win. Overall, aligned models seem to follow a common human behavior, to become more confident following a positive (or non-negative) outcome in a game, naturally pushing them to become more predictable.

## 4 Creative Poetry Generation

Finally, we test the complex challenge of being *creatively* unpredictable. Change and evolution are fundamental aspects of art (Fienberg & Martindale, 1991), meaning that the most impactful art must be novel and original i.e. not predictable. Following this intuition, we test the ability of models to be original in creative/artistic writing, specifically for poetry (Katz et al., 2023).

In a small-scale, contest-style human evaluation, we find that base models generate the most *original* poems in every case, while instruct models generate the most *pleasant* poems. Providing some intuition for this difference, we find that pleasantness is more strongly correlated with annotated human preference, a core aspect of many alignment techniques. Overall, our findings support the idea that aligned poetry is easy to read but less likely to be impactful or artistically interesting (Kestenbaum, 2024).

### 4.1 Experiments

**Setup** We prompt models to carry out a simple poetry exercise, generating fixed-length poems (4 lines) on a basic topic. We query GPT-4 (OpenAI et al., 2023) for a list of everyday topics: *Coffee, Smartphones, Traffic, Weather, Exercise, Grocery shopping, Sleep, Work, Internet, Television*. We then give models a straightforward, zero-shot prompt asking for a 4-line poem on the given topic, and generating until we are able to parse 20 poems of the given length. We find that 70B parameter models are significantly more proficient at this task, and so we focus our analysis on these models (using the same models as earlier sections).

| | **Human Eval Contest Wins** | | | **Example — Poems about Coffee** |
|---|---|---|---|---|
| | Original | Pleasant | Preference | |
| $\rho$ with **Preference:** | -0.08 | 0.34 | — | |
| **Base** | 4 | 0 | 1 | Adenosine wrapped in caffeine, ◁ 🏅 Original
Dance through my receptors within.
Stimulate; neurotransmitters go insane.
Kicked inside my synapses; wake up brain! |
| **Llama-Instruct** | 0 | 2 | 1 | In brewed aroma delight takes flight,
Awakening sleep with a gentle bite.
Caffeine weave through morning's space,
In every sip, the day we lace. |
| **Tulu-Full** | 0 | 2 | 2 | Rich aroma fills the air ◁ 🏅 Preference
As morning dawns, with coffee's care
Warm and dark, a soothing friend
Lifting fog, till the day's end. |

Figure 5: Results of creative poetry evaluation for 70B parameter models, where base models produce the most original, but not always preferred or pleasant, poems. *Left:* Number of wins along each axis for human evaluation poetry contests (§4.1), across 4 poem prompts, 5 poems per model, and 3 axes. We include average rank correlation ($\rho$) with the pleasantness axis across poems. *Right*: Examples from one contest, including the best-ranked poems by human preference and originality. Aligned poems seem to share a style that differs significantly from the base model. Further examples are included in Appendix A.3.2.

**Human Evaluation – Contest** Art is typically judged by the best rather than average case (e.g. in poetry contests), so we construct an evaluation to extract winners along different human-evaluated axes: *originality*, *pleasantness*, and *preference*. Originality serves as our notion of unpredictability/creativity, which is what our evaluation ultimately aims to test. Although annotator preference is often used as the measure of generation quality, there is no concrete evidence that this correlates well with broader artistic merit or impact, and our study finds that it correlates more with *pleasantness*, which may be at odds with novelty and impact.

To avoid leading annotators, we evaluate each of these axes separately in their own annotation tasks, comparing a series of random pairs of poems. For a given axis and set of poems, we determine a final winner by inducing an ordering using a variant of the Bradley-Terry model (Bradley & Terry, 1952) from the pairwise comparisons. We use this format to aggregate over natural disagreements in subjective questions, and so expect some disagreement between annotators. We also calculate annotator agreement with 70 additional comparisons, finding Cohen's Kappa (Cohen, 1960) of 0.33, 0.27, and 0.67 for originality, pleasantness, and overall preference (respectively).

We carry out human evaluation of the 2 most popular aligned models tested here (Tulu-Full and Llama-Instruct) along with the base model, all at 70B parameters. We evaluate the 3 axes for 4 different poem prompts (*coffee, sleep, weather, smartphones*), comparing 5 random poems from each model. This results in 12 contests of 15 poems each. We carry out 60 comparisons per contest on the Prolific platform, resulting in a total of 720 annotated comparisons. We also include one control question in every job, comparing between a generated poem and a random bag of words. Annotators selected the generated poem in virtually every case.

## 4.2 Results

**A split between preference and originality** The results of our human evaluation of model poetry are included in figure 5 (left). We include limited examples in figure 5 (right) and extensive examples in Appendix A.3.2. There is a distinct split in attributes: the base model produces the winning poem in terms of creative originality in all cases, but this does not translate to dominance in terms of human annotator preference. Indeed, when taking Spearman rank correlation (Spearman, 1904) averaged across contests, originality is actually slightly *negatively* correlated with human preference (mean $\rho$ across settings is $-0.08$). These results support our earlier findings that aligned models tend to be more predictable than base models, which results in a lower level of perceived artistic originality in this case.

**Pleasantness aligns with Preference** We find that the pleasantness axis aligns more positively with annotator preference in terms of rank (mean $\rho = 0.34$) than originality does (mean $\rho = -0.08$). Given that annotator preference is often a core element of alignment, their desire for pleasantness may explain why aligned models seem to prioritize this over originality. This also suggests that crowdsourced preference, commonly used as the ultimate test of quality in generations, may not give a clear a strong signal towards artistic originality or impact.

On the other hand, annotators seem to recognize the originality of the base model although they do not prefer it. The base model never wins in terms of pleasantness, and its poems also have the lowest median rank in terms of human annotator preference (10.5 in sets of 15 poems). Yet, the base model does win one of four contests in terms of preference, suggesting that annotator opinions of base model poems are highly variable.

## 5 Related Work

Our work aims to develop an understanding of broad model limitations and biases, particularly the **effects of alignment techniques**. Recent work has studied the relationship between base and aligned models, often focusing on the differences between them (Lin et al., 2024), and how to encourage aligned behavior (Hewitt et al., 2024; Fei et al., 2024). Like our work, Li et al. (2024) study the qualitative differences caused by alignment (particularly RLHF) and similarly find that alignment can narrow some capabilities not covered by task-based improvement. A growing body of work studies the loss of diversity that can result from alignment (Murthy et al., 2024; Kirk et al., 2023; Bronnec et al., 2024; Padmakumar & He, 2024), which is related to unpredictability studied here. Shypula et al. (2025) find the counterintuitive result that aligned models can have higher semantic diversity despite lower syntactic/surface diversity, particularly in settings where aligned models can generate more *high quality* answers. McCoy et al. (2023) also study the biased effects of model training techniques, although focus on issues induced by pretraining rather than post-training.

Other works attempt to measure writing creativity related to LLMs. Lu et al. (2025) measure this as the degree to which generations match existing text snippets, and similarly find that alignment (RLHF in that case) greatly reduces this notion of creativity. Many other works focus on the human experience of creativity, using human judgements instead Chakrabarty et al. (2024); Anderson et al. (2024); Gómez-Rodríguez & Williams (2023). Our work follows the latter approach. While our work focuses on direct model creativity, this is also relevant for the setting in which models collaborate creatively with humans on writing Padmakumar & He (2024); Chakrabarty et al. (2024); Anderson et al. (2024)

Multiple past works have studied the ability of models to carry out **random behavior**, such as random number generation (Hopkins et al., 2023; Bigelow et al., 2024; Koevering & Kleinberg, 2024), demographic sampling (Meister et al., 2024), or playing games that require randomness (Silva, 2025). None of these works aim to study the effects of alignment on randomness, although some observe an effect (Hopkins et al., 2023; Koevering & Kleinberg, 2024). Like our work, some study patterns in generated random sequences: Koevering & Kleinberg (2024) also find a tendency against repetition, while Bigelow et al. (2024) find models can transition from randomness to formal languages in different settings. Paruchuri et al. (2024) investigate the ability of models to reason about randomness rather than sample. More broadly, Song et al. (2025) advocate for evaluations that consider non-determinism more strongly.

One aspect of our work is studying model biases in settings that have strong human biases (randomness, mixed strategy games). Past work has studied these for a range of known human cognitive biases and opinions (Liu et al., 2025; Jones & Steinhardt, 2022; Itzhak et al., 2024; Santurkar et al., 2023).

Most works on random numbers look at binary (Bigelow et al., 2024; Koevering & Kleinberg, 2024) or continuous (Hopkins et al., 2023) distributions, while our experiments in this space use integer sampling. Other works study games for LLMs (Silva, 2025; Brookins & Debacker, 2023; Akata et al., 2023; Jia et al., 2025) but do not focus on the divide between aligned and base models as our experiments do.

## 6 Conclusion

Overall, our work provides extensive support to the notion that **popular alignment recipes erode a range of capabilities present in base models**. Despite better performance on common benchmarks, aligned models are found to have lower performance across a range of tasks tested here (figure 1).

Concretely, the alignment recipes studied here seem to reduce the ability of models to be *unpredictable*. This could have significant implications for the impacts of LLMs, given the dominance of aligned models. Practically, our findings could mean that aligned LLMs are not as effective at creative tasks, or assisting humans with creativity, given our results on poetry generation. Similarly, our results on games call into question how useful aligned models will be for settings requiring mixed strategies, like natural communication or bargaining on behalf of a user. On the other hand, this may have positive implications for safety, as aligned models may be less effective at deception, which is thought to require ambiguity and non-determinism.

One remaining question resulting from our work is whether there is an inherent tradeoff between unpredictability and the capabilities at which these aligned models excel. Exploring this question could shed light on the underlying mechanisms of model capabilities. Regardless, our work suggests that although base LLMs receive much less attention than their aligned forms, there are mysterious and valuable capabilities hidden within them.

## Ethics Statement

Our work carries out analysis of existing language models, and does not train any new models or introduce any new datasets. In all human evaluations carried out here, we follow necessary IRB guidelines, and aim to pay our workers $15 per hour on average.

One important point is that our work is advocating for the value of base language models, which could carry risks compared to aligned models. We would like to clarify that we only advocate for the deployment of safe systems to the general public. Our work does not imply that large and untested base models should be made available at large, but rather that current alignment techniques may erode useful capabilities that were available in the original base parameters.

## Acknowledgments

This work is supported in part by the Institute for Human-Centered AI at Stanford University. We thank Ari Holtzman, Jared Moore, and the Stanford NLP Group for useful input and feedback on this research.

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

# A    Appendix

## A.1    Experimental Details

## A.2    Overall

We use the suggested prompt formatting for each of the given models. These are:

**Tulu**

```
<|user|>
{instruction}
<|assistant|>
{optional infix}
```

where we may include an infix to aid in parsing, specifically observing intro text that Tulu includes typically includes before it returns an answer.

**Llama-Instruct:**

```
<|begin_of_text|><|start_header_id|>system<|end_header_id|>
You are an expert at following human instructions.
<|eot_id|><|start_header_id|>user<|end_header_id|>
{instruction}
<|eot_id|><|start_header_id|>assistant<|end_header_id|>
{optional infix}
```

With optional infix as defined above. Finally,

**Base:**

```
<|begin_of_text|>{instruction}
```

Note that the exact wording of the instructions may differ slightly for the base model, which often requires instructions to be framed more contextually than as a direct command.

**LLM Implementation**  We use the SGLang library for all language model inference. We use full precision for 8B parameter models on a single NVIDIA RTX A6000 GPU. For 70B parameter models, we use fp8 precision on 4 NVIDIA RTX A6000 GPUs. All models used here are based on the Llama 3.1 (Dubey et al., 2024) family of models.

### A.2.1    Games Requiring Randomness

**Adversarial Probability:**  We note here that the probabilities used to decide adversarial moves are approximate. We estimate the likelihood of each next model move by investigating the next-token distribution given the prompt: $p(t|prompt)$. For each token $t$ that corresponds to the beginning of a surface form for one of the given moves, we add this probability to the adversarial estimate of that move, and renormalize these combined probabilities in the end. Note that the actual move played by the model is decided by parsing model generations, which better indicates the underlying behavior of the model but does not allow for consistent probability estimation.

## A.3    Results

### A.3.1    Random Number Generation

**Adjusting for Entropy**  To test whether the superior performance of base models at generating single random numbers is simply an effect of their higher entropy distributions, we explicitly adjust for entropy in figure 7. In this experiments, we sample from each model at multiple temperatures (1.0, 1.5, 2.0, 2.5, 3.0). For each aligned model, we take the lowest temperature that gives a next-token distribution (following the prompt, i.e. $p(t|prompt)$) with an entropy at least as high as the base model. As the figure shows, the entropy in many cases is significantly higher than the base model, but this does not make these models

| model | 0 | 1 | 2 | 3 | 4 |
|---|---|---|---|---|---|
| Tulu DPO | 1774 | 2116 | 1461 | 1158 | 1377 |
| Tulu SFT | 983 | 1307 | 773 | 775 | 580 |
| Tulu Full | 1800 | 2241 | 1792 | 1248 | 1564 |
| Llama instruct | 9127 | 8839 | 7578 | 5800 | 7158 |
| Base | 107 | 120 | 145 | 164 | 128 |

Table 3: A comparison of multiple hand-written prompts for single random integer generation (comparable to figure 2). Values here are divergence, with paraphrased prompts numbered from 0 to 4.

as uniform as the base model. This indicates that the effect of higher entropy, rather than smoothing out the distribution, pushes more probability outside of the valid output space.

We further carry out a grid experiment across sampling hyperparameters ($top_p$, entropy) in tables 4 to understand the broader combined effects of these factors. Overall, we find that decoding parameters do not offer an easy way to make aligned models better at randomness. No tested settings allow aligned models to reach the default setting of the base model (temp = $top_p$ = 1.0), which achieves a divergence of 137. Note that these experiments are carried out on single random number generation (comparable to figure 2)

**Alternative model** In table 5 we include single random integer generation (comparable to figure 2) results for an alternative model architecture, the Qwen2.5 models (Qwen et al., 2025). We see patterns similar to our earlier results, with the aligned form seeing significantly larger divergence and no significant improvement with scale.

**Sampling of characters** We carry out an experiment that asks models to generate a random letter from the first 11 letters in the alphabet, rather than numbers 0-10 in the single integer generation experiment (figure 2). These results are presented in table 6, and show similar patterns to the case of integer generation. One significant difference is that the most commonly generated letter is highly dependent on model, alignment recipe, and scale. Holding scale constant, Tulu models agreed ("j" for 8B and "f" for 72B). This suggests that the alignment recipe has a significant impact on how aligned model models collapse.

**Sensitivity to prompting** We carry out a prompt analysis, testing 5 prompts for random integer generation with differing wording (paraphrased by hand). These results are included in table 3. While there is quite a bit of variation between performance with prompts, it is far from enough to push aligned models towards matching the performance of base models, and all models seem to follow similar prompt-wise patterns of divergence.

**Reward Model Analysis** We study the Tulu reward model (used for aligning some Tulu models) for the random integer generation task in table 7. Particularly, we get the reward for each random integer between 0 and 10 given the prompt for the original random integer generation experiment. We find similar rewards returned for many of these values. The integer that all Tulu models pick most frequently ("7") achieves the highest reward, by a very small amount. Yet, given the reward maximization inherent in many RL algorithms, an optimal policy model would maximize this reward by generating only 7, despite its very small advantage over other integers.

**Sequential random number generation histograms** We include histograms for sequential number generation (similar to figure 2) in figure 8. As discussed in the main paper, sequential generation results in more uniformity across models, but does not result in true randomness for the aligned models, which follow other heuristics (such as not repeating integers).

**Sequential random number generation – probability by position** In figure 6, we include 2D histograms of the probability of each integer being generated at each position in the 10-integer sequence. Note the relatively limited amount of structure in the Base and SFT histograms, while Llama Instruct, Tulu DPO, and Tulu Full seem to be highly conditional on position.

| temperature | top$_p$ | Base | Llama instruct | Tulu DPO | Tulu SFT | Tulu Full |
|---|---|---|---|---|---|---|
| 2.5 | 1.0 | 6 | 3028 | 467 | 149 | 649 |
|  | 0.8 | 7 | 3184 | 601 | 174 | 646 |
|  | 0.5 | 18 | 3275 | 818 | 211 | 1177 |
| 2.0 | 1.0 | 16 | 4242 | 774 | 356 | 826 |
|  | 0.8 | 19 | 4600 | 1023 | 353 | 1346 |
|  | 0.5 | 36 | 7623 | 2301 | 820 | 2402 |
| 1.5 | 1.0 | 56 | 5333 | 965 | 496 | 1055 |
|  | 0.8 | 40 | 8187 | 1725 | 896 | 1968 |
|  | 0.5 | 88 | 15000 | 3092 | 1883 | 3283 |
| 1.0 | 1.0 | 137 | 8324 | 1306 | 937 | 1602 |
|  | 0.8 | 258 | 10087 | 2233 | 1442 | 2250 |
|  | 0.5 | 1286 | 15000 | 3618 | 2074 | 8097 |

Table 4: A broad sweep of decoding hyperparameters: temperature and top$_p$. Note that the experimental setup here is comparable to figure 2.

| size | Base | Instruct |
|---|---|---|
| 7B | 792 | 10970 |
| 72B | 744 | 15000 |

Table 5: A repetition of the single integer generation experiment from figure 2, using the Qwen2.5 model rather than Llama.

### A.3.2   Poem Examples

We include example poems generated by models in table 8 and table 9 for the topics of coffee and sleep.

| size | Base | Instruct | Tulu DPO | Tulu SFT | Tulu Full |
|------|------|----------|----------|----------|-----------|
| 8B | 78 ("a") | 1158 ("e") | 614 ("j") | 370 ("j") | 832 ("j") |
| 70B | 193 ("k") | 10513 ("f") | 1177 ("f") | 709 ("f") | 1231 ("f") |

Table 6: Repeating the random integer experiment from figure 2, but prompting models to generate letters rather than numbers. Values are divergence, while the most frequently generated letter for each model is included in parentheses.

| Integer | 0 | 1 | 2 | 3 | 4 | 5 | 6 | 7 | 8 | 9 | 10 |
|---------|------|------|------|------|------|------|------|------|------|------|------|
| Reward | 3.98 | 4.37 | 4.99 | 5.15 | 5.19 | 5.03 | 5.26 | 5.29 | 5.28 | 5.01 | 4.23 |

Table 7: Reward values returned by the Tulu reward model, for various integers returned for random integer generation as in figure 2. This reward model is used in training some Tulu models. Note that "7" revieves the highest reward for this prompt, by a small margin.

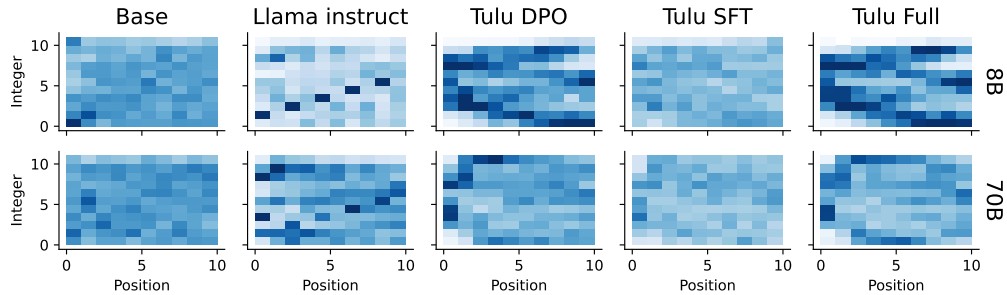

Figure 6: Plotting the probability of generating each integer at each position across models. Note that Base and SFT show relatively little structure (closer to uniform random) while all other aligned models show very high positional structure. Note that this is aggregated across sequential generations, and does not necessarily capture all probabilistic structure in each model, only structure that is highly position dependent (e.g. in Llama instruct)

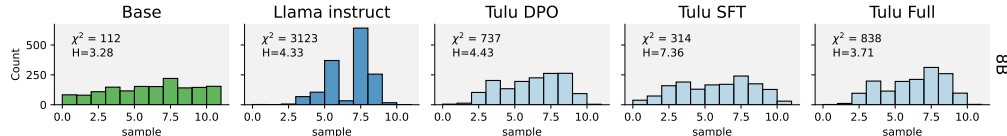

Figure 7: Histograms for single-value random integer generation, adjusting for entropy. We test temperatures of 1.0, 1.5, 2.0, 2.5, and 3.0, taking the lowest entropy for the aligned models where the resulting entropy (H in this plot) is at least as high as the base model (for 8B parameter models). Despite having higher overall token-wise entropy in each case, aligned models still have higher divergence ($\chi^2$) from uniform across the board.

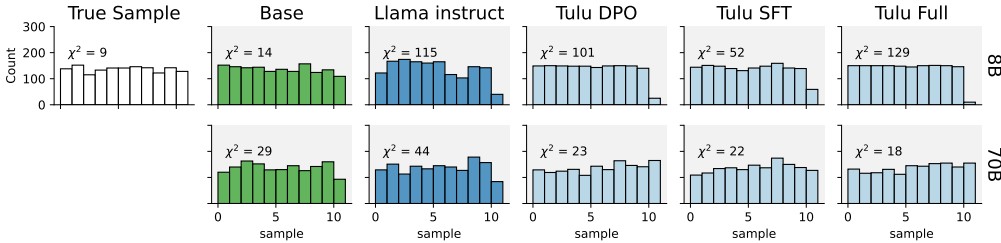

Figure 8: Analogous plot to figure 2 for integer sequences: histograms of sampled integer counts with divergence ($\chi^2$) from expected uniform counts across models and sizes.

| Base | Llama-Instruct |
|---|---|
| Mighty caffeine
rallying spirit
hot cups of comfort
midst bitter lives | Rich aroma fills the air
A morning pick-me-up beyond compare
Lifting fog from weary eyes
Invigorating senses, a warm surprise |
| Adenosine wrapped in caffeine,
Dance through my receptors within.
Stimulate; neurotransmitters go insane.
Kicked inside my synapses; wake up brain!
**\*Winner for Originality** | Steaming cup, aromatic delight
Lifting fog, and banishing the night
Rich flavors dance, a sensory thrill
Coffee's warmth, a morning's gentle will |
| Fresh brew softly sitting,
tempting each drinker
to add a drop of poison,
a bitter unrequited love. | Rich aroma fills the air
Lifting fog from morning's care
A cup of joe, a daily friend
Invigorating till the day's end |
| Enjoy the deep and dark flavors of this brew,
Savor the richness that's found in its hue.
Aromatic and smooth, a cup to cherish,
Coffee, our love, is our dear mistress. | Steaming cup, aromatic delight
Lifting fog from morning's earliest light
Rich flavors dance upon my tongue
Coffee's warmth, my daily song |
| Morning's wake up call,
Creamy, rich and bold,
Awakens the senses,
As each desire takes hold. | Rich aroma fills the air
As morning dawns, with coffee's care
Warm and dark, a soothing friend
Lifting fog, till the day's end.
**\*Winner for Pleasantness** |
| **Tulu-Full** | |
| In mugs of steam, the morning starts anew,
Sipping on dreams, bold and subdued.
Beans' aroma wakes the mind's embrace,
Coffee's spirit, in every sip, we trace. | |
| In brewed aroma delight takes flight,
Awakening sleep with a gentle bite.
Caffeine weave through morning's space,
In every sip, the day we lace. | |
| In the dark, a bean from distant lands,
Brewed to wake from morning's sweet demands.
The aroma swirls, a siren's call.
Awake to the world, embrace it all.
**\*Winner for Preference** | |
| Black elixir of the eager dawn,
Awakening the sleepy with its song.
In mugs and cups, its warmth is drawn,
And with each sip, the day is long. | |
| Aromatic brew to start the day,
Infusing warmth with every sip;
Dark elixir, complex, bold, and gay,
Waking spirits, quicken, energize, flip. | |

Table 8: Poems from various 70B models on the topic of coffee, along with best poems by axis as judged by humans.

| Base | Llama-Instruct |
|---|---|
| Sleep is a temporary death. 
 A little rest, 
 I lift my eyelids 
 and behold the world's vastness. | Softly darkness wraps the night 
 As the world succumbs to sleep's delight 
 Dreams dance in the silent brain 
 Restful slumber soothes the strain |
| Sweet sleep, how soothing is thy receding power, 
 When the world's cares assail me in the hour 
 Of loneliness and longing; and the place 
 Lies dark and desolate before my face. 
 **\*Winner for Originality** | Softly falls the evening night 
 As tired eyes succumb to sleep's sweet light 
 In tranquil rest, the body heals 
 And weary souls their vigor reveals 
 **\*Winner for Pleasantness** |
| Restful repose amid quiet night, 
 Dreams laid to rest, souls taking flight, 
 Bridging the gap twixt heaven and hell, 
 Scarce a single thought to tell. | Softly darkness folds around my mind 
 Sleep's sweet respite, where worries unwind 
 In slumber's realm, I find my peaceful nest 
 And dreams transport me to a world at rest |
| Teach me to be gentle, to love my spring days. 
 Teach me to be kind and unhard, to sleep. 
 To rest in flowery night-beds, to fruit, 
 To live human, to die. | Softly darkness falls around, 
 Sleep's sweet solace is found. 
 In slumber, worries cease to be, 
 Restful dreams set the soul free. |
| Tip-toe, feet, 
 Quiet, it's bedtime, 
 and peaceful dreams are 
 waiting to be slept. | Softly darkness falls at night 
 Sleep's sweet solace calms the fight 
 Weary eyes, now closed in rest 
 Dreams unfold, and souls are blessed 
 **\*Winner for Preference** |
| **Tulu-Full** | |
| In sleep, we find a haven deep and still, 
 A realm where night transcends to fulfill. 
 Dreams' fleeting scenes, a chorus to fulfill, 
 Our slumber's voyage, a nightly drill. | |
| In land of dreams, Sleep gently taps, 
 Whispering tales from moon's soft glow, 
 With closed eyelids, the mind unwraps, 
 Secrets beneath the shroud of woe. | |
| In the quiet cradle of night, dreams unfold, 
 A realm where shadows and stars are closely knold. 
 In slumber's gentle hands, the world lies still, 
 While souls wander freely, both far and until. | |
| In realms of soft repose at night we lay, 
 Eyes heavy, shut by dreams' gentle sway. 
 Time forgets to race; the restless mind slows, 
 In sleep's embrace, worries and cares decompose. | |
| Silent watcher of our deepest dreams, 
 Where worries fold and weighted lids descend, 
 A fleeting realm of whispers and moonbeams, 
 Sleep, restorer of our tired life's trend. | |

Table 9: Poems from various 70B models on the topic of sleep, along with best poems by axis as judged by humans.

