# OpenReview forum: "Base Models Beat Aligned Models at Randomness and Creativity"
_colmweb.org/COLM/2025/Conference — COLM 2025_

### Official Review · Reviewer_wJrW · 2025-05-05

**Rating:** 7
**Confidence:** 4
**Ethics Flag:** 1

**Summary:**

The main claim of this paper is that alignment of LLMs through techniques such as reinforcement learning from human feedback has a negative impact on their ability to perform taks that require unpredictable outputs. This claim is supported by a number of experiments comparing aligned versions of models from the Llama-3.1 to their unaligned base models on tasks such as random number generation, mixed-strategy games like rock-paper-scissors, and poetry generation. The experiments show that the unaligned base models consistently outperform their aligned counterparts with respect to randomness and/or originality. The experimental evidence is convincing and further supported by additional analyses in several cases. The main limitation of the paper is that the experiments are restricted to two base models from the same model family, which makes it hard to assess the generalizability of the results.

**Questions To Authors:**

Did you consider running experiments besides Llama-3.1. If yes, what made you decide against it?

In an otherwise superbly clear paper, I find the following sentence a little convoluted: "This suggest that the usefulness of alignment is not the only intuition that may breakdown in tasks requiring unpredictability." What exactly do you want to say here? Should "breakdown" possibly be "break down"?

Typos and such:
- Section 4.1: Extra space after "Work" and "Internet" in the first paragraph.
- Section 5: I think "[good term?]" should be removed.
- Ethics Statement: Missing negation in "does train any new models" (I assume).

**Reasons To Accept:**

- Interesting findings that expand our knowledge of LLMs and associated training techniques.
- Solid experiments and good analysis.
- Very clearly written.

**Reasons To Reject:**

- The main weakness of the paper is the restriction of experiments to a single model family (in two sizes), but I would not really say that this is a reason to reject.

---

> ### Author Response · Authors · 2025-06-03
>
> We thank the reviewer for finding that our work “expands knowledge of LLMs” and is “very clearly written”! We address questions and comments below:
>
> ### **restriction of experiments to single model family**
>
> At the request of multiple reviewers, we have carried out experiments on Qwen-2.5 models. At roughly comparable sizes, we see similar results to Llama, for example, seeing more than 10X increase in divergence after alignment on random integer generation (see Table A below). We note that the base Qwen performance is significantly worse than base Llama, but the effect of alignment seems the same. We will incorporate results on Qwen in the final version, and are happy to add other models as resources allow.
>
> ```bash
> | size   |   Base |   Instruct |
> |:-------|-------:|-----------:|
> | 7B     |    792 |      10970 |
> | 72B    |    744 |      15000 |
> ```
>
> **Table A:** Divergence ($\chi^2$) of random integer sampling with Qwen 2.5 models (directly comparable to Figure 2 in the paper, lower is better)
>
> ### **confusing sentence “… that may breakdown… ” and typos**
>
> Thank you for noticing this typo — this is indeed supposed to be “break down”, and the intended meaning is that the common intuition of improvement through scale also breaks down in some of our experiments.
>
> Thank you for noticing the other typos in our work. These (and a number of typos we also noticed later) will be fixed in our final version!

---

> > ### Comment · Reviewer_wJrW · 2025-06-04
> > **Thanks**
> >
> > Thanks for sharing experimental results with a different model family.

---

> > > ### Author Response · Authors · 2025-06-10
> > >
> > > We thank the reviewer for engaging with our response

---

### Official Review · Reviewer_u55E · 2025-05-12

**Rating:** 6
**Confidence:** 3
**Ethics Flag:** 1

**Summary:**

This paper investigates how alignment procedures (e.g., Supervised Fine-Tuning, DPO, RLHF) affect the ability of large language models to exhibit randomness and creativity. Using a set of designed experiments across three domainsL random number generation, mixed-strategy games (Rock-Paper-Scissors, Hide-and-Seek), and creative poetry generation. They find that base (unaligned) models consistently outperform aligned models on tasks requiring unpredictability or novelty. The aligned models display deterministic tendencies (e.g., overusing the number "7"), reduced strategic diversity, and a preference for pleasant but less original poetry. These findings shows a previously underexplored trade-off that alignment techniques that improve benchmark scores and user satisfaction may degrade other important capabilities such as stochastic reasoning and creative expression.

**Questions To Authors:**

Could decoding strategies (e.g., temperature, top-p) recover lost randomness in aligned models? Paper mention using temperature 1.0 with full top-k sampling, but don’t show how aligned models perform under varied decoding settings. Have you tried tuning sampling parameters to see if aligned models can approximate base model behavior in these tasks? Would increasing entropy during decoding help mitigate deterministic tendencies introduced by alignment?

The paper focuses on short-form tasks. Would the trade-off between alignment and unpredictability also appear in Storytelling tasks requiring long-range narrative structure? or Agents in multi-agent environments requiring adaptive strategies?

Can you share some insights or intuition that why do you think alignment procedures introduce deterministic behavior?  Do you have logit-level analyses showing whether alignment procedures sharpen output distributions? Could alignment be reinforcing “safe” or majority responses (e.g., “7”) due to reward shaping or over-regularization?

**Reasons To Accept:**

- This paper addresses a critical yet underexplored topic that the unintended consequences of alignment procedures (e.g., RLHF, SFT, DPO) on model unpredictability and creativity. As most public-facing LLMs are aligned, understanding the trade-offs involved is of central importance to both the research community and industry deployment.
- The authors conduct a methodologically rigorous evaluation across three domains including Random number generation, Mixed-strategy games (Rock-Paper-Scissors, Hide-and-Seek) and Creative writing (poetry).
- The paper shows a consistent trade-off that as models become more aligned and perform better on standard NLP benchmarks, they exhibit reduced randomness and originality in domains where these qualities are crucial.

**Reasons To Reject:**

- The paper repeatedly refers to "randomness" and "unpredictability" but does not clearly define what constitutes sufficient or task-appropriate randomness. For example, the authors evaluate model outputs using $χ^2$ divergence and repetition rates, but they do not evaluate whether the model’s behavior is strategically optimal (e.g., ε-Nash in games) or just different from uniform randomness.
- The three tasks such as uniform integer generation, Rock-Paper-Scissors, and 4-line poem generation, are limited proxies for real-world unpredictability and creativity. While conceptually appealing, they suffer from oversimplification.
- In Appendix A.3.1, the authors claim that increasing entropy through temperature tuning still leads to less randomness in aligned models, but the evidence is presented in a cursory way.  For example, Figure 7 in the appendix shows entropy-adjusted comparisons, but the models' sampling space and decoding parameters (e.g., top-k, top-p) are not discussed in detail, nor is the interplay between model logits and sampling strategy interrogated.
- The human evaluation for poetry relies on just 720 pairwise annotations (60 per contest over 12 contests) and uses Prolific crowdworkers. There is no validation of inter-annotator agreement, task clarity, or demographic diversity.
- While the paper compute useful divergence metrics (e.g., χ², MSE, entropy), the paper does not conduct a meaningful error analysis. There are no examples of failure cases, no category-level breakdown of errors, and no investigation into why specific alignment techniques cause predictable behavior. For example, The models are shown to overproduce the number “7,” but no introspection is done into whether this arises from instruction phrasing, training data artifacts, or logit suppression. Maybe would be better to Include qualitative examples of   failures case (e.g., repetitive poems, exploitable strategies in games) and conduct a diagnostic analysis of what parts of the alignment process cause these.

---

> ### Author Response · Authors · 2025-06-03
>
> We thank the reviewer for recognizing that our work “addresses a critical yet underexplored topic”. We address comments and questions below:
>
> ### **Proxies vs Real-world**
>
> We agree that our tasks do not reflect all complex aspects of the real world. We made this design choice for 2 reasons. First, such tasks tend to incorporate many different capabilities (unpredictability, coherence, world knowledge, etc.) making it difficult to isolate and evaluate individual aspects such as unpredictability. Using proxy tasks allows us to define settings with one central capability and clear evaluation metrics (such as $\chi^2$). Second, given the failure of strong aligned models on even these simple tasks, it seems unlikely that they can be effectively unpredictable in more complex settings. As abilities improve, more complex evaluations will be required, yet simple tasks serve as an effective first hurdle.
>
> ### **Task-appropriate randomness**
>
> We thank the reviewer for pointing out that more detail would be useful in this part of the work—we are happy to add a discussion of ε-equilibrium. For example, in the case of rock paper scissors, expected payoff for a uniform strategy is 0 points regardless of the adversary. All models achieve a payoff of at most -15 points, meaning all have significant incentive to unilaterally deviate to a uniform strategy (≥15), indicating non-ε-equilibrium for small ε. We are happy to expand on this however it might be useful.
>
> ### **Human evaluation concerns**
>
> First, we note that the scale of the study, including pilots, already accounts for 100s of dollars at a reasonable pay rate, which is a limiting factor. We will still attempt to increase the scale by a reasonable amount.
>
> We have calculated Cohen's Kappa with ~70 additional comparisons, finding 0.33, 0.27, and 0.67 for originality, pleasantness, and overall preference (respectively). Our original goal in this evaluation was to use contests to find consensus under possible disagreement, given the subjective nature of the task and past findings that agreement as a notion of quality overlooks genuine sources of disagreement [2, 3, 4]. We also note that we included one control question in each job, comparing a generated poem to one composed of randomly selected words from other poems. Annotators chose the generated poem in the vast majority of cases. We will include the full templates, results of control experiments, and agreement information in the final version.
>
> ### **Error analysis and causes**
> We are happy to include more examples and analysis of error cases, such as repetitive poems generated by aligned models (see examples below), comparing these effects across model sizes and alignment recipes. We will also extend our analysis of model biases in games, e.g., building on our observation in Figure 4 that aligned models tend to become more confident following ties/wins vs. losses. We will include new experiments studying the factors affecting breakdown of randomness. For example, we run random letter generation (Table B below, similar to Figure 2 but generating letters “a” to “k”) and find that aligned models still collapse onto peaks, but the top letters differ between alignment recipes and model sizes. This suggests that collapse onto single answers is not simply the result of very likely tokens, as the preference of models for “7” may have suggested.
>
> ### **Entropy analysis concerns, and “Could decoding strategies … recover lost randomness?”**
> First, we would like to clarify that all sampling parameters (outside of Appendix A.3.1) are set for pure sampling i.e. no truncation (top-p/k) and a temperature of 1.0. We aimed to directly compare distributions, avoiding confounding factors. The reviewer is correct that sampling parameters can have an effect, although out of top-p/k and temperature, only temperature would *increase* randomness/entropy. Appendix A.3.1 suggests that even relatively high values of temperature do not allow aligned models to match randomness of base. In the limiting case (very high temperature), aligned models *would* become uniform but also incoherent, once all tokens have the same probability. We will include the full results of our temperature grid, where even very high temperatures (which greatly decrease the quality of aligned models) failed to match the uniformity of base models.
>
> ### **Longer form tasks**
> Although we have not explicitly tested this, there is likely a family of long-form tasks (e.g. artistic/creative storytelling) where both base and aligned models will often fail—aligned models will not be original enough, while base models will not be structured enough. We believe there is an exciting space of hybrid solutions that combine base and aligned models, although this is beyond the scope of our work here. Overall, this highlights the fact that more complex tasks make it harder to isolate individual capabilities for evaluation, one motivation for keeping our tasks relatively simple.

---

> > ### Author Response · Authors · 2025-06-03
> >
> > ### **insights into why alignment introduces deterministic behavior?**
> >
> > Past work, such as [1] supports the tendency of alignment techniques to “sharpen output distributions” at the logit level, as the reviewer suggests. We believe that reward likely plays a role here, as the alignment techniques more explicitly incorporating reward (RLHF, DPO) were found to result in more determinism than the case without it (supervised finetuning). Intuitively, reward creates some notion of a “best” output, which may result in peaks when optimized. Yet, this issue could also be a general feature of adapting models towards specific tasks or behaviors, as suggested in [1]. The fact that Qwen base is less random than Llama, while including more task-oriented synthetic data in pertaining, supports this idea. We will provide more discussion in the final version.
> >
> > ### **Poetry examples (topic=coffee, LLM=Llama 70B)**
> >
> > **Base**
> >
> > (1)
> >
> > Mighty caffeine
> >
> > rallying spirit
> >
> > hot cups of comfort
> >
> > midst bitter lives
> >
> > (2)
> >
> > Adenosine wrapped in caffeine,
> >
> > Dance through my receptors within.
> >
> > Stimulate; neurotransmitters go insane.
> >
> > Kicked inside my synapses; wake up brain!
> >
> > (3)
> >
> > Fresh brew softly sitting,
> >
> > tempting each drinker
> >
> > to add a drop of poison,
> >
> > a bitter unrequited love.
> >
> > **Instruct**
> >
> > (1)
> >
> > Rich aroma fills the air
> >
> > A morning pick-me-up beyond compare
> >
> > Lifting fog from weary eyes
> >
> > Invigorating senses, a warm surprise
> >
> > (2)
> >
> > Rich aroma fills the air
> >
> > As morning dawns, with coffee’s care
> >
> > Warm and dark, a soothing friend
> >
> > Lifting fog, till the day’s end.
> >
> > (3)
> >
> > Rich aroma fills the air
> >
> > A morning pick-me-up beyond compare
> >
> > Lifting fog from weary eyes
> >
> > Invigorating senses, a warm surprise
> >
> > ### **Tables**
> >
> > ```bash
> > | size   |   Base |   Instruct |
> > |:-------|-------:|-----------:|
> > | 7B     |    792 |      10970 |
> > | 72B    |    744 |      15000 |
> > ```
> >
> > **Table A:** Divergence ($\chi^2$) of random integer sampling with Qwen 2.5 models (directly comparable to Figure 2 in the paper, lower is better)
> >
> > ```
> > | size   | Base      | Instruct    | Tulu DPO   | Tulu SFT   | Tulu Full   |
> > |:-------|:----------|:------------|:-----------|:-----------|:------------|
> > | 8B     | 78 ("a")  | 1158 ("e")  | 614 ("j")  | 370 ("j")  | 832 ("j")   |
> > | 70B    | 193 ("k") | 10513 ("f") | 1177 ("f") | 709 ("f")  | 1231 ("f")  |
> > ```
> >
> > **Table B:** Divergence ($\chi^2$) for random letter generation (letters a to k) with Llama-based models (directly comparable to Figure 2, lower is better). Results are listed as:  $\chi^2$ (*most likely letter*)
> >
> > ### **References**
> >
> > [1] Li, Margaret et al. “Predicting vs. Acting: A Trade-off Between World Modeling & Agent Modeling.” *ArXiv* abs/2407.02446 (2024): n. pag.
> >
> > [2] Amidei, Jacopo et al. “Rethinking the Agreement in Human Evaluation Tasks.” *International Conference on Computational Linguistics* (2018).
> >
> > [3] Liu, Alisa et al. “We're Afraid
> > Language Models Aren't Modeling Ambiguity.” *ArXiv* abs/2304.14399 (2023): n. pag.
> >
> > [4] Basile, Valerio et al. “We Need to Consider
> > Disagreement in Evaluation.” *Proceedings of the 1st Workshop on Benchmarking: Past, Present and Future* (2021): n. pag.

---

> > > ### Comment · Reviewer_u55E · 2025-06-07
> > >
> > > Thanks for the detailed reply and the new examples. The clarifications helped, especially on ε-equilibrium, poetry controls, and error examples.
> > >
> > > A few points are still unclear or need more work:
> > > - The reply mention high temperatures don’t help, but it’s still not clear how decoding settings (like top-p, temperature >1.0) affect aligned models in detail. Showing the full sweep would help.
> > > - The reply suggests reward sharpens outputs, but there’s no clear analysis of how or where this happens (e.g., logits, entropy over training). More concrete evidence would be useful.
> > > - It’s still not clear if the randomness measured is task-appropriate (e.g., strategic in games vs. just token-level entropy). This could be better explained or evaluated.
> > > - The reply propose to add examples and discuss certain repetitive tendencies (e.g., number “7,” letter “f”), which is appreciated. However, the response still lacks causal analysis, e.g., what features of RLHF or DPO (reward design, data used, optimization dynamics) are driving this determinism beyond sharp distributions? Are there logit-level shifts, entropy decay over training steps, or reinforcement of high-likelihood outputs?

---

> > > > ### Author Response · Authors · 2025-06-10
> > > >
> > > > We thank the reviewer for continuing to engage deeply with our work and response.
> > > >
> > > > **Decoding settings?** We include a sweep of decoding parameters below (**Table C**), presenting $\chi^2$ divergence for random integer sampling on 8B parameter models. As the reviewer requested, we sweep values for temperature and top_p together. A first observation is that no aligned model matches pure sampling (temp,top_p=1.0) with the base model, which achieves  $\chi^2$=137. Tulu SFT gets closest at temp=2.5, top_p=1.0, with $\chi^2$=149. However as we stated above, this results in highly incoherent behavior, with a success rate for generating valid random integers 30X lower than the base model with pure sampling (only about 3% of generations are valid). In other words, high temperature can result in uniformity for aligned models, but also vastly lowers the quality of the output distribution. We will include more comprehensive outputs for this sweep in the final version, including full histograms in the appendix.
> > > >
> > > > **What causes sharpening?** We conduct a new experiment to study the role of rewards in this sharpening. Below (**Table D**), we calculate reward with the Tulu-3 reward model on all output options for random integer generation. As expected, “7” achieves the highest reward, although “6” and “8” have very similar reward values. Any model that purely aims to maximize reward would be incentivized to generate only “7” in this case, despite only a slight preference in reward, which could help explain the behavior seen in our experiments. We will expand on this in the final verison.
> > > >
> > > > **Task-appropriateness:** We will attempt to clarify this point, and ask the reviewer to correct any misunderstanding we may have about this concern. One point to clarify is that our primary metric for games is model success within the game, not an intrinsic measure of randomness. We assert that this measure would be task appropriate given that it is direct game performance. Its connection to randomness is that this is a measure of *worst-case* or *adversarial* performance, in which randomness is required to achieve a high score and thus a high score directly implies randomness.
> > > >
> > > > The reviewer may also wonder whether aligned models have a poor worst-case performance in these games, but are still strategically effective against typical opponents (e.g. a human). We do not explicitly test this case here, but argue that this strategic success would be less tied to randomness, by definition, as it deviates from success in our setting which explicitly requires maximum uniform randomness. Strategies that specifically predict opponent moves can be effective, but will be more deterministic and adversarially vulnerable. We will clarify all of these points in the final version.
> > > >
> > > > **What features of RLHF or DPO are driving this?** Our experiment on the Tulu reward model (discussed above) gives some initial intuition that reward design/definition plays a role. We agree that studying this over training steps is essential, and we are actively running experiments on publicly available Tulu post-training checkpoints to answer this question. Given the computational cost of analyzing a full training trajectory across task settings, this was not feasible to complete before the end of the discussion period. We believe that causal explanations here will fundamentally require interventions on the post-training process itself, which would be significantly more resource-intensive and warrants its own study.

---

> > > > > ### Author Response · Authors · 2025-06-10
> > > > >
> > > > > ```
> > > > > |   temperature |   top_p |   Base |   Llama instruct |   Tulu DPO |   Tulu SFT |   Tulu Full |
> > > > > |--------------:|--------:|-------:|-----------------:|-----------:|-----------:|------------:|
> > > > > |           2.5 |     1   |      6 |             3028 |        467 |        149 |         649 |
> > > > > |           2.5 |     0.8 |      7 |             3184 |        601 |        174 |         646 |
> > > > > |           2.5 |     0.5 |     18 |             3275 |        818 |        211 |        1177 |
> > > > > |--------------:|--------:|-------:|-----------------:|-----------:|-----------:|------------:|
> > > > > |           2   |     1   |     16 |             4242 |        774 |        356 |         826 |
> > > > > |           2   |     0.8 |     19 |             4600 |       1023 |        353 |        1346 |
> > > > > |           2   |     0.5 |     36 |             7623 |       2301 |        820 |        2402 |
> > > > > |--------------:|--------:|-------:|-----------------:|-----------:|-----------:|------------:|
> > > > > |           1.5 |     1   |     56 |             5333 |        965 |        496 |        1055 |
> > > > > |           1.5 |     0.8 |     40 |             8187 |       1725 |        896 |        1968 |
> > > > > |           1.5 |     0.5 |     88 |            15000 |       3092 |       1883 |        3283 |
> > > > > |--------------:|--------:|-------:|-----------------:|-----------:|-----------:|------------:|
> > > > > |           1   |     1   |    137 |             8324 |       1306 |        937 |        1602 |
> > > > > |           1   |     0.8 |    258 |            10087 |       2233 |       1442 |        2250 |
> > > > > |           1   |     0.5 |   1286 |            15000 |       3618 |       2074 |        8097 |
> > > > > ```
> > > > > **Table C:** Sweep of decoding parameters (temperature, top_p), with $\chi^2$ divergence for 8B parameter models on random integer generation (comparable to Figure 2).
> > > > >
> > > > > ```
> > > > > |    0 |    1 |    2 |    3 |    4 |    5 |    6 |    7 |    8 |    9 |   10
> > > > > |-----:|-----:|-----:|-----:|-----:|-----:|-----:|-----:|-----:|-----:|-----:
> > > > > | 3.98 | 4.37 | 4.99 | 5.15 | 5.19 | 5.03 | 5.26 | 5.29 | 5.28 | 5.01 | 4.23
> > > > > ```
> > > > > **Table D:** Output of the Tulu 3 reward model on all valid outputs for random integer generation (0 to 10). Returning “7” achieves the highest reward, although it is similar in value to “6” and “8”.

---

### Official Review · Reviewer_ze1Q · 2025-05-24

**Rating:** 8
**Confidence:** 4
**Ethics Flag:** 1

**Summary:**

The paper discusses the issue of model alignment and provides a family of tasks on which base (unaligned) large language models (LLMs) perform better than aligned models. The tasks include problems that require unpredictable outputs or decisions, or creativity  (random number generation, mixed-strategy games, and poetry generation). On each of these tasks, they present a series of evaluations comparing base LLMs vs. aligned LLMs (they use Llama-3.1 as the base model and its finetuned variants. They show that on each task, the base model outperforms the aligned variant. For the poetry generation task, they conduct a human study showing that a base model always performs better with respect to creativity, but it does not translate to dominance in terms of human preference.

**Reasons To Accept:**

This is an interesting paper that provides evidence that popular alignment recipes erode a range of capabilities present in base models, specifically, they reduce the ability of the models to be unpredictable.

**Reasons To Reject:**

NA

---

> ### Author Response · Authors · 2025-06-03
>
> We thank the reviewer for positive comments! We are happy to address any concerns as they arise.

---

### Official Review · Reviewer_beTY · 2025-05-26

**Rating:** 6
**Confidence:** 4
**Ethics Flag:** 1

**Summary:**

This paper studies the effect of alignment on the predicitibility of outputs for random number generation and games such as rock-paper-scissors and hide-and-seek. The authors find that base models have much more diverse generations, when compared to the aligned models which tend to concentrate around certain values, leading to almost deterministic and predictible outputs. The authors confirm their findings empirically by prompting the model the generate a large amount of numbers and checking how close the distribution is to a uniform, or by trying to adversarially come up with strategies exploiting the deterministic nature of the aligned models.
Finally, the authors also show that in tasks regarding generating poetry, base models are generally the most original ones.

**Questions To Authors:**

Is there any way to quantify the impact of the pretraining data on the pattern of the generations? I believe that there must be a significant difference between models pretrained on different datasets.

In Table 2, it seems that Tulu-SFT is almost consistent second best for the net outcome. Do you have any speculation why this might be the case, as opposed to any of the other methods?

**Reasons To Accept:**

The paper studies an interesting question regarding the impact of aligning models on the generations produced by the models. The authors study both quantitatively this effect (by measuring how far the distribution of "random" numbers generated by a model is from an actual random uniform sample, as well as in games such as rock-paper-scissors and hide-and-seek by seeing how much the determinism can be exploited in post-trained models), as well as qualitatively (in tasks such as creative generation).

The authors do an extensive ablation over the alignment procedures and scales, studying the base Llama model (across 8B and 70B scale), aligned with the Llama-Instruct, SFT, DPO and both. The results interestingly show that scale does not alleviate the issue of diversity collapse in models, but that the different training protocols can lead to a difference in the generation diversity.

The authors also investigate the task of creative writing, showing that, consistently with the previous tasks, the base models have more originality in their generations than the aligned models.

**Reasons To Reject:**

The authors only report their findings on a single subset of models, namely Llama models. While I understand the constraints due to compute reasons, many works have shown that there is very high variance between the performance and behaviour of different model families such as Qwen, Tulu etc.

Whilte the findings reported by the authors are mostly consistent across the games they test on, these games are very simple and have few states and actions. It would have been interesting to see if a similar deterministic pattern is maintained for games with longer time horizons, such as chess, Go.

---

> ### Author Response · Authors · 2025-06-03
>
> We thank the reviewer for positive comments and careful feedback. We respond to comments and questions below:
>
> ### **different model families such as Qwen**
>
> As requested, we ran the Qwen model family, and broadly see results in line with our previous findings. For example, for random integer generation at comparable sizes to Llama (Table A below), Qwen also sees a large jump in divergence after alignment, despite the base model already being significantly more divergent than Llama. We will include results for Qwen in the final version and are happy to add other models as resources allow.
>
> ```bash
> | size   |   Base |   Instruct |
> |:-------|-------:|-----------:|
> | 7B     |    792 |      10970 |
> | 72B    |    744 |      15000 |
> ```
>
> **Table A:** Divergence ($\chi^2$) of random integer sampling with Qwen 2.5 models (directly comparable to Figure 2 in the paper, lower is better)
>
> ### **Games with Longer Time Horizons**
>
> We agree that these games (such as chess and Go) would be interesting. The main challenge is that the role of randomness is much less clear/harder to evaluate. For example, chess endgames (the final few moves) can often be solved totally deterministically by the winner, and therefore unpredictability may actually be a negative. A main goal in our work was providing explicit evaluations in settings where the benefit of unpredictability is clear, and so this may be out of our scope. However, we believe a study of when and how models may use unpredictability in these games would be extremely valuable.
>
> ### **How to quantify the impact of pretraining data on the pattern of generations?**
>
> A first step in understanding the impact of pretraining data is studying models with different pretraining. For example, added experiments on Qwen models (Table A, above) show that pretraining may have an impact on initial unpredictability, as Qwen base is significantly more divergent from uniform than similarly-sized Llama. This may indicate that the inclusion of synthetic data during pretraining, as used in Qwen, has a significant effect.
>
> ### **Tulu-SFT is consistently second best**
>
> While a definitive explanation will need more evidence, we have some ideas for why this may happen. First, this may relate to post-training objectives. Other aligned models explicitly incorporate reward with methods like RLHF and DPO, which could push models towards a highest-reward output and cause peaks. In contrast, SFT uses the same cross-entropy objective as pretraining which works to learn the underlying data distribution. At a higher level, this may be related to a general trade-off between unpredictability and popular tasks like those in the Open LLM Leaderboard. SFT does not push models as far as other alignment techniques towards solving such Leaderboards, but this may also mean it does not trade off as much ability to be unpredictable.

---

> > ### Comment · Reviewer_beTY · 2025-06-04
> > **Reply**
> >
> > Thank you very much for the replies!
> >
> > ## different model families such as Qwen
> > The new results on Qwen are much appreciated! These strengthen the claims of the paper and I would like to thank the authors for adding them. As an additional comment here, I think future work in characterizing the precise effect of the pretraining data on the diversity of the output would be really valuable.
> >
> > ## Games with Longer Time Horizons
> > I understand and agree with your point regarding chess and Go. However, I still believe that testing on games with a longer time horizon where being unpredictible helps would be interesting. I believe that Poker would actually qualify for such a game, as it entails multiple turns, and being predictible makes you susceptible to the adversary consistenly winning against you over multiple rounds.
> >
> > ## How to quantify the impact of pretraining data on the pattern of generations?
> > See comment I wrote on the first point.
> >
> > ## Tulu-SFT is consistently second best
> > Thank you for this explanation. Intuitively, I believe your argument makes sense. To my understanding, what you are saying is that SFT aims to match the distribution of responses (which is diverse), whereas the RLHF algorithms are collapsing this distribution on the modes that are "most helpful"/preferred by humans. It would be interesting to see either in future work or in further versions of this paper an experiment confirming this hypothesis.
> >
> > In general, I believe that this is a good paper with a clean and easy to understand hypothesis and answer, and I would suggest acceptance. From a scientific curiosity point of view, I would be very happy if the authors could expand the points I (alongside the authors in this discussion) raised in a future version of the paper, or in future work.

---

> > > ### Author Response · Authors · 2025-06-10
> > >
> > > We thank the reviewer for responding to our comments in detail, and for saying that they would **suggest acceptance!**
> > >
> > > We are glad that the results on **Qwen** are useful, and strongly agree that understanding the effect of pretraining is vital. We are actively looking into this for future work, starting with comparisons between different pretraining recipes, as with our results on Qwen and Llama. Still, we believe a study controlling for pretraining factors will eventually be needed.
> > >
> > > We also agree that **poker** would be an excellent game for longer time horizons. The vulnerability of deterministic strategies implies that an adversarial setup similar to our games experiments would work. One significant complexity is the partially observable environment. This would mean more interesting and rich behavior from models, but could be complex enough to justify its own future study.
> > >
> > > Finally, your interpretation of our theory about **SFT** is correct, and we share the belief that confirming this will be valuable. To study whether reward plays a role, we add one more experiment testing the Tulu reward model on random integer generation below (Table B). As predicted, the highest reward goes to “7”. While “6” and “8” achieve similar rewards **any model that is maximizing this reward would still be pushed to generate “7” only**. This supports the idea that reward maximization is a factor, and we will expand on this study in the final version.
> > >
> > > We believe that stronger proof will come from studying extensive post-training checkpoints released for Tulu, although these experiments were not feasible to run before the end of the discussion period.
> > > ```
> > > | 0    | 1    | 2    | 3    | 4    | 5    | 6    | 7    | 8    | 9    | 10   |
> > > |:-----|:-----|:-----|:-----|:-----|:-----|:-----|:-----|:-----|:-----|:-----|
> > > | 3.98 | 4.37 | 4.99 | 5.15 | 5.19 | 5.03 | 5.26 | 5.29 | 5.28 | 5.01 | 4.23 |
> > > ```
> > > **Table B:** Output of the Tulu 3 reward model on all valid output options for random integer generation (0 to 10). “7” achieves the highest reward, although it is similar in value to “6” and “8”.

---

### Official Review · Reviewer_D7L6 · 2025-05-26

**Rating:** 6
**Confidence:** 4
**Ethics Flag:** 1

**Summary:**

This work studies the potential negative effects of alignment post-training techniques in domains which require unpredictability and creativity. The authors argue that although alignment methods such as RLHF improve safety and instruction-following capabilities, they can significantly degrade capabilities like randomness and creative expression. They verify this on a random number generation (single number and sequence generation), mixed strategy games (rock paper scissors and hide-and-seek), and creative poetry writing. For evaluating the creative writing, humans graded according to originality, pleasantness, and preference; the authors note a positive correlation between pleasantness and human preference, but a negative correlation between originality and preference. Base models consistently won in originality and aligned models were preferred for being more pleasant. Overall, the authors claim that there is a systematic trade-off between benchmark performance (improved via alignment) and the ability to behave unpredictably or creatively.

**Questions To Authors:**

* Have you tried evaluating other base models? I mention Qwen (and I know this is a low-hanging fruit to ask as a reviewer) but my reason for this is that Qwen 'base' models are markedly distinct from Llama in that it underwent synthetic-data-heavy pretraining. I'm curious if the 'base' models there will already exhibit the same issues as the instruction-tuned models here.
* Given the simplicity of your tasks, I believe it would also be meaningful to include comparisons to models where the objective has specifically included a term to encourage diversity [1], or at least highlight the potential i.e. to serve as a litmus test for assessing unpredictability and originality.
* Does manipulating the prompt to ask explicitly-- either for originality or for letting the model know optimal play in light of the adversary-- help aligned models perform better on these tasks?
*  Do you believe the observed drop in randomness can be attributed to phenomena that can be observed during training or in the model internals (eg. rank collapse)?

[1] Lanchantin, Jack, et al. "Diverse Preference Optimization." arXiv preprint arXiv:2501.18101 (2025).

**Reasons To Accept:**

* The paper addresses an important consequence of alignment—its tendency to reduce unpredictability and creativity in language models.
* The experimental design is compelling, with a clean set of synthetic tasks to exemplify loss of models' ability to produce random outputs that has been anecdotally observed or computed using various diversity metrics (eg. n-gram based)
* Results are consistently in favor of base models, with use of reasonable quantitative metrics and human evaluations to back the claims
* The results as alignment methods are applied to larger models is particularly interesting, since it has been noted that truly consistent instruction-following occurs around the 70B-scale.

**Reasons To Reject:**

(See questions for further elaboration)
* The novelty is perhaps limited, given multiple existing works which cite the loss of diversity at the cost of optimizing for safe outputs [1]
* In particular, while this work identifies a loss in aligned model's ability to produce unpredictable outputs/generate original creative writing, it stops short of proposing solutions or mitigations. As the authors have observed, adjusting temperature or accounting for entropy doesn’t help because you want models to `act random' within a certain set of restrictions. More demanding creative writing formats like storytelling would test a model’s ability to maintain coherence alongside originality, which I suspect base models to perform worse than aligned models.
* The study is restricted to a single family (LLaMA-3.1-based) of models.
* The work would benefit from having a more extensive literature review on works which study how alignment methods amplify human cognitive biases (to name a few, [2,3,4])

[1] Kirk, Robert, et al. "Understanding the effects of rlhf on llm generalisation and diversity." arXiv preprint arXiv:2310.06452 (2023).

[2] Murthy, Sonia K., Tomer Ullman, and Jennifer Hu. "One fish, two fish, but not the whole sea: Alignment reduces language models' conceptual diversity." arXiv preprint arXiv:2411.04427 (2024).

[3] Itzhak, Itay, et al. "Instructed to bias: Instruction-tuned language models exhibit emergent cognitive bias." Transactions of the Association for Computational Linguistics 12 (2024): 771-785.

[4] Santurkar, Shibani, et al. "Whose opinions do language models reflect?." International Conference on Machine Learning. PMLR, 2023.

---

> ### Author Response · Authors · 2025-06-03
>
> We are happy to hear the reviewer finds our “experimental design is compelling” and our work “addresses an important consequence of alignment”. We respond to reviewer questions and comments below:
>
> ### **“other base models? … Qwen”**
>
> As suggested, we replicated experiments with Qwen-2.5. Results support both our findings that alignment reduces unpredictability, and the reviewer’s idea that the Qwen base model “already exhibit[s] the same issues”. For example, in random integer generation (Table A below), base Qwen models at comparable sizes to Llama are significantly more divergent (e.g. $\chi^2$ of 744 for Qwen vs 355 for Llama at ~70B parameters). Alignment still results in a large (more than 10X) increase in divergence.
>
> We agree that including DivPO (Lanchantin et al. 2025) would be valuable. Model weights do not seem to be available yet, but we will add this once they are. We predict that DivPO will result in performance closer to base models in Figure 1, with strong unpredictability but weaker performance than other aligned models on the OpenLLM Leaderboard.
>
> ```bash
> | size   |   Base |   Instruct |
> |:-------|-------:|-----------:|
> | 7B     |    792 |      10970 |
> | 72B    |    744 |      15000 |
> ```
>
> **Table A:** Divergence ($\chi^2$) of random integer sampling with Qwen 2.5 models (directly comparable to Figure 2 in the paper, lower is better)
>
> ### **Novelty vs. existing works on diversity**
>
> While past work on linguistic diversity is relevant, there is not a clear optimal level of diversity that models should have, making evaluation less definitive. One way our work is distinct is in proposing tasks where success is clearly defined, e.g., total uniformity in random integer generation.
>
> ### **Solutions or mitigation, more demanding formats**
>
> The central goal of our work was to develop a deeper understanding and more explicit evaluation for these effects of alignment, but we can suggest possible solutions. We believe the two most promising directions are enhanced alignment techniques that more explicitly handle this tradeoff (e.g. Lanchantin et al. 2025, which you mention), and creating hybrid systems that allow the strengths of base and aligned systems to work together.
>
> This is related to your suggestion of “more demanding creative writing formats like storytelling”. We agree that base models may fail due to a lack of coherence, while aligned models would fail due to a lack of originality. A system that allows the two to work together may solve this. We will discuss further in the updated version.
>
> ### **suggested citations on amplified cognitive bias**
>
> We thank the reviewer for the suggested citations. We will add these along with other citations such as [1,2], which also deal with cognitive biases and aligned models.
>
> ### **Does manipulating the prompt help?**
>
> While models are sensitive to prompting, this does not seem to be a solution. For random number generation, the prompt is very clear that randomness is needed, but the issue persists. For games, we did experiment with prompt versions that encode the optimal strategy, yet there tended to be a ceiling on randomness for the aligned models. Base models showed some brittleness and could be less random for certain prompts, but saw consistently better performance and a much higher ceiling. We will add this study to the final version.
>
> ### **Causes: Training and Model Internals**
>
> Given the consistency of this effect, we believe it is quite likely that it would be observable during post-training, likely including telltale changes in model weights. A natural next step would be pairing the evaluations we developed with iterative post-training checkpoints (e.g. OLMo 2 which has weights at every 20 training steps) to observe when and how these issues arise. We also believe that the objective function may have an important effect, given that SFT models (which use cross entropy-based probabilistic learning) seemed to fare better than models that explicitly incorporate reward. This may require a more comprehensive study, post-training multiple models with different objectives.
>
> ### **References**
>
> [1] Liu, Ryan et al. “Large Language Models Assume People are More Rational than We Really are.” ICLR (2025)
>
> [2] Jones, Erik and Steinhardt, Jacob  “Capturing Failures of Large Language Models via Human Cognitive Biases.” NeurIPS (2022).

---

> > ### Comment · Reviewer_D7L6 · 2025-06-06
> >
> > Thank you to the authors for addressing my questions, and running additional experiments to include Qwen results during the rebuttal period. I am sympathetic to the author's approach in identifying tasks which attempt to establish well-defined settings that evaluate a model's creative ability, which is known to be difficult. If the authors add these additional results, the additional references, and the study on prompt manipulations, I lean towards accepting the paper.

---

> > ### Author Response · Authors · 2025-06-10
> >
> > We thank the reviewer for fully considering our response. We are very happy that they **lean towards accepting our work** given these additions to the paper.
> >
> > We include a sample of the prompt manipulation experiments below (Table B), using 8B models for the random number generation task. Supporting our earlier statement, prompting does affect aligned models, but they seem to have a ceiling in terms of randomness. For example, prompt 4 uses very strong phrasing (explicitly reiterating that “all integers have equal probability”), but aligned models are still far less uniform than the base model. We will include similar analysis across model sizes and tasks with diverse sets of prompts, along with the previously described additions to the final paper.
> >
> > | model          |    0 |    1 |    2 |    3 |    4 |
> > |:---------------|-----:|-----:|-----:|-----:|-----:|
> > | Tulu DPO       | 1774 | 2116 | 1461 | 1158 | 1377 |
> > | Tulu SFT       |  983 | 1307 |  773 |  775 |  580 |
> > | Tulu Full      | 1800 | 2241 | 1792 | 1248 | 1564 |
> > | Llama instruct | 9127 | 8839 | 7578 | 5800 | 7158 |
> > | Base           |  107 |  120 |  145 |  164 |  128 |
> > **Table B:** $\chi^2$ divergence for random integer generation (comparable to Figure 2 in the paper) for 8B parameter models across multiple prompts. Prompts are numbered 0 to 4, with some highlighting the need for randomness (e.g. prompt 4) and others leaving the phrasing more subtle.

---

### Official Review · Reviewer_bHor · 2025-05-27

**Rating:** 5
**Confidence:** 4
**Ethics Flag:** 1

**Summary:**

The authors contend (and show with some evidence) that fine-tuning through RLHF reduces the randomness and creativity of generation, pushing it more towards a singular/less uniform outcome distribution. The authors test this on three tasks: (i) generating a single random number and generating a sequence of 10 random numbers, (ii) mixed strategy games, (iii) poem generation. In tasks (i) and (ii) the authors show that the fine-tuned models for the most part exhibit a less uniform distribution over outcomes/game choices than the base model, even when the optimal strategy would be unpredictability. For part (iii), the authors show that base model poems are often more original vs fine-tuned models through human evaluation.

**Questions To Authors:**

Most of my reasons to reject are also things that I believe, if changed, could significantly improve the quality of the paper. In addition,

1. What about if you don't use numbers, but some other set of tokens, and ask it to randomly generate one of a set of tokens? Does this show the same effect?
2. Is the entire token distribution more peaked (perhaps suggested by (iii))? One interesting thing would be to look at the perplexity of base model generations under the fine-tuned models, and vice versa.
3. It's know that you can estimate a biased coin with a von-neumann extractor. Is there a similar analog to this when using aligned language models that could be analogous?
4. Given this discovery, what takeaways should the reader have? (It seems a bit unclear how the reader should use these results)

**Reasons To Accept:**

This paper is clear and easy to understand. Further, I believe that the poem generation experiments are worthwhile and important for general knowledge. This problem also seems to be important (as the authors motivate) as people interact more frequently with aligned models. Further, the authors make a good attempt to provide details for reproducibility of their experiments.

**Reasons To Reject:**

There are a few concerns I have with the experiment fundamentals of experiments (i) and (ii):
1. Since experiments are done with open models, I would think that looking the logits to observe a probability distribution would be more meaningful than running a number of samples.
2. There isn't any coverage of sampling parameters, which also could play a role in the generation tasks
3. I am also concerned that using a single class of model may not fully capture this phenomenon because of different tokenization methods (maybe ablation over these differences would be helpful)

---

> ### Author Response · Authors · 2025-06-03
>
> We are happy that the reviewer found our work “clear and easy to understand” and our experiments “important for general knowledge”. We address comments and questions below:
>
>
> ### **“What takeaways should the reader have?”**
>
> One high-level takeaway from our work is that aligned models do not always represent the limits of current model capabilities. While they are by far the most popular, base models may succeed on tasks for which aligned models fail, which readers should consider when selecting models. A second, related takeaway is that more work is needed to understand the effects of alignment. As part of this, our work supports the idea of broader evaluations for models, especially evaluations for which single, static responses are not enough. Finally, new methods are needed to combine the benefits of aligned and base models.
>
>
> ### **Samples vs logits**
>
> When designing our experiments, our main concern with using logits was that these require picking an exact prefix to the response token, while both base and aligned models often vary generation format for a given prompt (e.g., adding “\n”, “Here is my answer:” etc. after the input prompt). Sampling allows the model to generate naturally across different formats from which answers can be extracted, while using logits would constrain the model to one format. However, recorded logits throughout, and generally saw the same patterns, meaning this distinction may not be too significant. We include a comparison in the final paper.
>
> ### **Role of Sampling Parameters**
>
> We use pure sampling in our main experiments (no truncation, temperature=1.0) to directly compare model distributions and avoid confounding variables. We agree that parameters would have an effect, although truncation (top-p/k) will only reduce entropy (i.e. make models even more predictable). We did experiment with temperature in Appendix A.3.1, although even when raising the aligned model entropy to match the base model, the aligned model was still more predictable at random number generation. We will include a more extensive grid of temperatures and further analysis in the final paper.
>
> ### **Single class of model**
>
> At the request of multiple reviewers, we produced results for the popular Qwen family of models as well, at comparable sizes to Llama. We see similar effects of alignment, for example, on random number generation (Table A below). Note that base Qwen is significantly more divergent than base Llama at similar sizes, possibly due to the synthetic task data included in pretraining. Still, alignment results in a more than 10X increase in divergence at both model sizes. We are happy to extend to more models as resources allow, and will incorporate these results in the final paper.
>
> ```bash
> | size   |   Base |   Instruct |
> |:-------|-------:|-----------:|
> | 7B     |    792 |      10970 |
> | 72B    |    744 |      15000 |
> ```
>
> **Table A:** Divergence ($\chi^2$) of random integer sampling with Qwen 2.5 models (directly comparable to Figure 2 in the paper, lower is better)
>
> ### **Random sampling of tokens instead of numbers**
>
> We ran this experiment, asking for uniform samples of letters from “a” to “k” (analogous to Figure 2, which uses integers 0-10). We see a similar overall pattern, with alignment making divergence significantly worse. While “7” was the most likely number across models, the most likely letter depends on both model and size—all 8B Tulu models prefer “j”, while all 70B aligned models prefer “f”. This suggests that convergence towards a mode is not simply the result of an overwhelming bias across models (as “7” may have suggested). We will include these results and further analysis in the paper—thank you for the very informative suggestion!
>
> ```
> | size   | Base      | Instruct    | Tulu DPO   | Tulu SFT   | Tulu Full   |
> |:-------|:----------|:------------|:-----------|:-----------|:------------|
> | 8B     | 78 ("a")  | 1158 ("e")  | 614 ("j")  | 370 ("j")  | 832 ("j")   |
> | 70B    | 193 ("k") | 10513 ("f") | 1177 ("f") | 709 ("f")  | 1231 ("f")  |
> ```
>
> **Table B:** Divergence ($\chi^2$) for random letter generation with Llama-based models (directly comparable to Figure 2, lower is better). Results are listed as:  $\chi^2$ (*most likely letter*)

---

> > ### Author Response · Authors · 2025-06-03
> >
> > ### **Entire distribution is more peaked?**
> >
> > Aligned token distributions are typically much more peaked. Past works such as [1] support this, showing that aligned models (RLHF in that case) give very high probability to their own generations, and mostly negligible probability outside of that narrow space. This means much worse perplexity for anything it didn’t generate, even the original pretraining distribution, which base models capture well. In our experiments, we tended to see:
> >
> > PPL(base | aligned) > PPL(base | base) > PPL(aligned | base) > PPL(aligned | aligned)
> >
> > where PPL(X|Y) indicates perplexity of generations from model X using model Y, i.e., the highest perplexity case is using aligned probability on base generations. We will add this analysis to the final paper.
> >
> > ### **Von Neumann Extractor for aligned LLMs?**
> >
> > This is a great point, you could absolutely use something like a Von Neumann extractor to get an unbiased distribution from the aligned models. This would solve problems in the first experiments (number generation and mixed strategy games). However, this becomes nontrivial when we don’t have the target distribution, as in the poetry writing experiments. The simple version of this can solve the surface problems, yet not the deeper problem. The potential for analogous concepts to extractors for notions such as linguistic diversity is exciting, but left for future work.
> >
> > ### **References**
> >
> > [1] Li, Margaret et al. “Predicting vs. Acting: A Trade-off Between World Modeling & Agent Modeling.” *ArXiv* abs/2407.02446 (2024): n. pag.

---

> > > ### Comment · Reviewer_bHor · 2025-06-05
> > >
> > > I thank the authors for their work and the detailed response to my questions.
> > >
> > > I think for the most part, this response alleviates my concerns. With respect to the part "“What takeaways should the reader have?”, I was more suggesting that the authors include a section in the paper which makes this a bit more clear. But I leave that to their judgment.
> > >
> > > For the logit comparison, I think it would also be instructive to see logit distribution from multiple prefixes. However, I believe the point that for testing only end result, you want the model to generate by itself.
> > >
> > > I think the authors indicate that they will change the paper significantly -- and in a direction which I think reflects a much stronger paper. I will increase my score likewise, but I cannot directly recommend acceptance as the changes are indeed so significant.

---

> > > > ### Author Response · Authors · 2025-06-10
> > > >
> > > > We thank the reviewer for carefully engaging with our response and raising the score.
> > > >
> > > > We agree that making the takeaways clearer for the reader would improve the paper, and we plan to do this in a new section for the final paper version, including the takeaways we mentioned above. Finally, we will add the comparison of logits and generation to the final version, with a focus on testing across multiple prefixes. Our experiments so far indicate that these differences are minimal, but we plan to expand the diversity of prefixes tested.

---

### Decision · Program_Chairs · 2025-07-08

**Decision:**

Accept

**Comment:**

This paper examines issues related to the alignment of LLMs. In particular, RLHF has emerged as a popular technique for aligning LLMs; however, RLHF and similar techniques have their limitations. Specifically, when a task requires a degree of unpredictability, aligned models tend to exhibit narrower behavior and perform worse than their baseline model counterparts.

**Reasons to Accept:**
- The paper is clear, easy to understand, and tackles an important problem in alignment.
- The experimental setting is compelling and thoroughly executed. Specifically, the authors conduct an extensive ablation study of the alignment procedure and its scaling effects.

**Reasons to Reject:**
- The paper identifies an issue with loss-aligned models but does not propose a solution to mitigate it.
- It does not thoroughly explain what is meant by randomness and unpredictability.
- The human evaluation of poems relies on 720 pairwise annotations, but there is a lack of inter-annotator agreement.